# Targeting the Deterministic Evolutionary Trajectories of Clear Cell Renal Cell Carcinoma

**DOI:** 10.3390/cancers12113300

**Published:** 2020-11-09

**Authors:** Adam Kowalewski, Marek Zdrenka, Dariusz Grzanka, Łukasz Szylberg

**Affiliations:** 1Department of Clinical Pathomorphology, Collegium Medicum in Bydgoszcz, Nicolaus Copernicus University in Torun, 85-067 Bydgoszcz, Poland; d_grzanka@cm.umk.pl (D.G.); l.szylberg@cm.umk.pl (Ł.S.); 2Department of Tumor Pathology and Pathomorphology, Oncology Centre-Prof. Franciszek Łukaszczyk Memorial Hospital, 85-796 Bydgoszcz, Poland; marek.zdrenka@cm.umk.pl

**Keywords:** clear cell renal cell carcinoma, ccRCC, RCC, kidney cancer, evolution, evolutionary trajectory, biomarker

## Abstract

**Simple Summary:**

In contrast to organismal evolution, human cancers are subjected to similar initial conditions and follow a limited range of possible evolutionary trajectories. Therefore, the repetitive nature of cancer evolution may prove to be its greatest weakness. Evolutionary trajectories of clear cell renal cell carcinoma (ccRCC) have been recently described. In this review, we will discuss the relevance of estimating the trajectory of ccRCC evolution as a readout for a response to therapy. Next, we will propose strategies to take advantage of the evolving nature of these tumors for patients’ benefit.

**Abstract:**

The emergence of clinical resistance to currently available systemic therapies forces us to rethink our approach to clear cell renal cell carcinoma (ccRCC). The ability to influence ccRCC evolution by inhibiting processes that propel it or manipulating its course may be an adequate strategy. There are seven deterministic evolutionary trajectories of ccRCC, which correlate with clinical phenotypes. We suspect that each trajectory has its own unique weaknesses that could be exploited. In this review, we have summarized recent advances in the treatment of ccRCC and demonstrated how to improve systemic therapies from the evolutionary perspective. Since there are only a few evolutionary trajectories in ccRCC, it appears feasible to use them as potential biomarkers for guiding intervention and surveillance. We believe that the presented patient stratification could help predict future steps of malignant progression, thereby informing optimal and personalized clinical decisions.

## 1. Introduction

Renal cell carcinoma (RCC) is the eighth most commonly diagnosed cancer in the United States, with an estimated incidence of 74,000 new cases in 2020 [1]. The classic triad of flank pain, flank mass, and hematuria occurs only in 10% of cases [2]. Due to the ability of the kidney for functional compensation when part of it is destroyed, early detection from loss of function is usually impossible. As a result, RCC remains clinically occult for most of its course, and around one-third of patients present with metastatic disease at the time of diagnosis. Those with localized tumors have up to 40% risk of recurrence following complete resection [3,4]. Remarkable advances over the last decade contributed to the development of targeted therapies and immunotherapies that today represent a standard for unresectable RCC. Despite relatively high response rates to these agents, the vast majority of patients eventually experience cancer progression. The emergence of clinical resistance to currently available systemic therapies represents a significant challenge and forces us to rethink our approach to RCC.

The best-studied histological subtype is clear cell renal cell carcinoma (ccRCC), which is derived from the proximal convoluted tubule and accounts for approximately 70% of all cases [5]. A series of next-generation sequencing studies led to a better understanding of the genetic background of ccRCC [6,7,8,9,10]. The results of these studies uncovered a near-universal inactivation of the von Hippel-Lindau disease (*VHL*) tumor suppressor gene. Other frequent alterations involve histone-modifying genes, SWI/SNF complex, and PI3K/AKT/mTOR pathway. Moreover, an integrated, genome-wide analysis of copy-number changes and gene expression profiles in ccRCC identified 7 chromosomal regions of recurrent arm level or focal amplifications (1q, 2q, 5q, 7q, 8q, 12p, and 20q) and 7 regions of losses (1p, 3p, 4q, 6q, 8p, 9p, and 14q) [8].

The evolutionary landscape in ccRCC is dominated by intratumor heterogeneity (ITH) at a genetic, transcriptomic, and functional level [9]. The exome sequencing performed on multiple, spatially separate ccRCC samples revealed that that two-thirds of the somatic mutations are not shared between all the primary tumor regions [10]. Hence, single-biopsy analysis is likely to miss the key genetic events or misclassify them as clonal. Apart from the direct impact on diagnostic procedures and biomarkers development, ITH has significantly hindered our understanding of ccRCC evolution.

In comparison to other malignancies, ccRCC is characterized by a high prevalence of somatic copy number alterations (SCNAs) and a low burden of somatic substitutions [6,8,11,12]. The integrative analysis of the genetic and clinical data led to the identification of certain alterations with prognostic value, such as mutually exclusive mutations of *BAP1* and *PBRM1* [13,14,15]. These studies, although conducted on large cohorts of patients, did not determine the prognostic values of genetic alterations according to whether they were clonal or subclonal. Huang et al. were among the first to demonstrate the possibility of genomic subtyping of ccRCC [13]. Recently, Turajlic and colleagues provided a comprehensive model of ccRCC evolution [14], which might lay the foundation for the development of precision clinical management.

Cancer cells continuously undergo adaptive changes, and insensitivity to drugs arises due to genetic and epigenetic alterations that offer a survival advantage. While there is a number of pathways and networks a cancer cell has at its disposal, targeting individual components is likely to prove inadequate [15]. Instead, the ability to influence cancer evolution itself by inhibiting processes that propel it or manipulating its course might potentially put an end to cancer as a major health concern.

In this review, we will discuss the relevance of estimating the trajectory of ccRCC evolution as a readout for a response to therapy. Next, we will propose strategies to take advantage of the evolving nature of these tumors for patients’ benefit.

## 2. The Origin, Evolution, and Routes to Metastasis of Clear Cell Renal Cell Carcinoma

### 2.1. The Origin of Clear Cell Renal Cell Carcinoma

Loss of the short arm of chromosome 3 is a nearly universal driver of ccRCC [16]. It occurs in childhood or adolescence, predominantly through chromothripsis. The deleted region encompasses at least four tumor suppressor genes, including *VHL*, *PBRM1*, *BAP1*, and *SETD2.* This earliest event produces a pool of a few hundred cells, which after decades of modest clonal expansion, acquire the necessary additional genetic alterations [17]. Chromosomal copies of deleted suppressor genes are often affected afterward, with inactivation of the second allele of *VHL* being the most common (65–80% of patients) [7,8,10]. In some cases, there are different driver mutations on the trunk of the phylogenetic tree, which, in contrast to 3p loss and *VHL* inactivation, trigger a substantial expansion [11,18].

### 2.2. The Evolutionary Trajectories of Clear Cell Renal Cell Carcinoma

On the basis of mutational ordering, timing, and co-occurrence, ccRCCs are classified into seven distinct evolutionary subtypes, or four groups, which correlate with clinical phenotypes [17,19]. These groups are distinguished by four features—variations in chromosomal complexity, ITH, model of tumor evolution, and metastatic potential. The variations in chromosomal complexity are measured as the fraction of the genome affected by SCNAs and expressed as a weighted genome instability index (wGII). ITH is measured as the ratio of subclonal drivers to clonal drivers [20].

Group 1 consists of primary tumors with *VHL* alteration as the sole driver event. They evolve in a “linear” fashion and are characterized by low both wGII and ITH. This mode of evolution is associated with indolent growth and low metastatic potential. Group 2 includes tumors in which early *PBRM1* mutation and subsequent *SETD2* mutation or PI3K pathway mutation or acquisition of SCNAs result in a “branched” evolutionary pattern. These are heterogeneous neoplasms with oligometastatic potential and attenuated progression. Clonal acquisition of multiple driver mutations (*VHL* plus ≥2 *BAP1*, *PBRM1, SETD2*, or *PTEN*) or the parallel *BAP1* mutation results in “punctuated” evolution. These tumors are characterized by high wGII but low ITH and belong to group 3. Punctuated evolution, driven mostly by high wGII, leads to rapid dissemination and is also observed among *VHL* wild-type tumors, which constitute the fourth group [14].

### 2.3. The Routes to Metastasis of Clear Cell Renal Cell Carcinoma

Metastasis competence is afforded by chromosome-level alterations that simultaneously affect the expression of hundreds of genes. These alterations provide a permissive genomic background for the selection of hallmark drivers of ccRCC metastasis and the loss of 9p and 14q [20]. Linear and branched evolution modes are analogous to Darwin’s phyletic gradualism. On the other hand, punctuated evolution, as in punctuated equilibrium, is associated with rapid speciation events and considerable evolutionary changes. Thus, the acquisition of metastatic competence is far more likely through punctuated evolution.

## 3. Current Systemic Therapies for Renal Cell Carcinoma

Immunotherapy and/or tyrosine kinase inhibitors (TKI) constitute the standard of care for relapse or stage IV RCC. Appropriate clinical management depends on disease activity, according to the National Comprehensive Cancer Network (NCCN) Guidelines for Kidney Cancer. In favorable-risk patients, first-line treatments include a combination of axitinib plus pembrolizumab or monotherapy with pazopanib or sunitinib. For patients with poor- and intermediate-risk disease, the preferred regimen is ipilimumab with nivolumab or axitinib with pembrolizumab. Moreover, cabozantinib may be considered in a first-line setting, especially in cases with osseous metastatic RCC. Because of the significant toxicity of systemic therapies, a subset of asymptomatic patients with metastatic RCC may benefit from active surveillance.

A major advantage of immunotherapy is its potential to produce complete and durable responses in a subset of patients with advanced cancer, even after discontinuation of the drug. Indeed, despite the non-curative nature of systemic therapy in RCC, up to 9% of poor- and intermediate-risk patients may achieve a complete response, according to the results of subgroup analysis of CheckMate 214 clinical trial [18]. This rate could be further increased by introducing novel treatment modalities as well as better patient selection algorithms.

## 4. Strategies to Overcome the Evolution of Renal Cell Carcinoma

In the face of selective pressures, subpopulations of tumor cells with adaptive phenotypes emerge at the expense of others. The ability to predict the alterations in ITH along the temporal axis seems invaluable for the development of personalized therapy. In this section, we will provide a summary of recent strategies against RCC which, when contextualized within an evolutionary framework, could be significantly more effective.

### 4.1. Cytoreductive Nephrectomy

In select patients with metastatic RCC, primary nephrectomy is performed with cytoreductive intent. Apart from the alleviation of symptoms associated with larger masses, such intervention eliminates the reservoir of phenotypic tumor-cell diversity, minimizing the risk of further metastatic seeding from an evolving primary tumor [19]. While cytoreductive nephrectomy (CN) is associated with a significant risk of perioperative mortality (0–13%) and major complications (3–36%) [21], there is a great need to avoid unnecessary surgery in nonresponders.

Heng et al. examined the role of CN in metastatic RCC patients receiving targeted therapies in a retrospective study of data from the International Metastatic Renal Cell Carcinoma Database Consortium (IMDC). They found that patients with estimated overall survival (OS) of <12 months and those exhibiting fewer than 4 IMDC prognostic factors are not likely to benefit from CN [22]. From that time, several other observational studies demonstrated analogous results [23]. This data, however, must be treated with caution given the significant risk of selection bias inherent to their study designs, which potentially leads to misclassification of patients [24].

The role of CN continues to change amid a rapidly increasing armamentarium of systemic therapies. In the modern immuno-oncology era, CN is still a viable option, but careful patient selection is of paramount importance. The ongoing clinical trials are evaluating the use of deferred CN in patients receiving nivolumab and ipilimumab alone or alongside radiotherapy (NCT03977571, NCT04090710). These studies may help determine the most appropriate indications for CN.

### 4.2. Adaptive Therapy

In the case of disseminated cancer with no significant probability of cure, patient survival can be maximized if adaptive therapy is introduced. This strategy originates from mathematical models and aims at maintaining a stable tumor burden [25]. When drugs are administered sparingly and in a temporally dynamic fashion, a significant population of treatment-sensitive cells survives. These, due to their competitive advantage, suppress the proliferation of treatment-resistant populations under normal tumor conditions.

Adaptive therapy may play a role in metastatic RCC. Findings from a prospective phase II trial demonstrate active surveillance to be a viable initial strategy in patients with few adverse prognostic features [26]. Results from the SURTIME study, a randomized clinical trial comparing immediate vs deferred CN, revealed that deferred CN is a valid option for patients with the intermediate-risk disease and with general clinical conditions at baseline amenable to undergo surgery [27]. “Treatment-for-stability” may also be represented by an alternative schedule of sunitinib. The standard dosing schedule of sunitinib is 50 mg daily for 4 weeks, followed by 2 weeks off drug (schedule 4/2). However, according to a recent meta-analysis, the administration of sunitinib for 2 weeks followed by 1 week off (schedule 2/1) exhibited lower toxicity and lower rates of treatment discontinuation while maintaining comparable responses [28].

The full potential of adaptive therapy is yet to be witnessed. Frequency-dependent game-theoretic models of tumor evolution have enabled the introduction of three concepts to consider in the pursuit of designing a multi-drug adaptive approach [29]. These ideas focus on entrapping tumor evolution in periodic loops, limiting the evolutionary “absorbing region” reachable by the tumor and determining the optimal timing of drug administration. Each may contribute to the generation of new treatment schedules and comparisons to standards.

### 4.3. Targeting Trunk Mutations

The ability to target alteration present in all tumor cells is expected to diminish the odds of the escape of clonal branches. As previously described, inactivation of *VHL* constitutes the trunk event in ccRCC development while most of the other driver aberrations are subclonal. Apart from large chromosomal aberrations as in the cytogenetic 3p abnormalities, *VHL* inactivation may be caused by small deletions affecting the locus, or promoter methylation and epigenetic silencing [30]. pVHL, a *VHL* gene product, is essential in the cell’s normal response to ischemic stress. Decreased expression of *VHL* results in the accumulation of hypoxia-inducible factor alpha (HIFα). Among the three known HIFα subunits, HIF2α is thought to be the core ccRCC driver since it upregulates a series of hypoxia-responsive genes [31,32,33]. The net effect is the activation of various kinase-dependent signaling pathways, such as MAPK/ERK and PI3K/AKT/mTOR [34]. While the most significant targets of *VHL* loss are the production of VEGF and PDGF, HIF2α has been regarded as undruggable for years [35,36]. Eventually, a structure-based design approach led to the identification of PT2385, a first-in-class HIF2α antagonist [37]. In a phase I dose-escalation clinical trial, PT2385 was found to be well-tolerated and demonstrated clinical activity in extensively pretreated ccRCC patients [38]. Its efficacy and safety are currently being evaluated in a phase II trial (NCT03108066). The primary objective of this trial is to assess the overall response rate in patients with VHL disease-associated ccRCC.

According to the mathematical model presented by Bozic et al., in the case of metastatic disease, monotherapy with a targeted agent offers no hope for recovery. Instead, combinations of two or more agents given simultaneously offer a small chance of cure, especially in the absence of cross-resistance mutations [39].

It is worth noting that the aforementioned drugs are directed against downstream effectors of *VHL*, hence, from an evolutionary point of view, there is a potential to better define the molecular target. Nicholson et al. found that inhibiting the cyclin-dependent kinases CDK4 and CDK6 impaired tumor growth in *VHL*-deficient ccRCC regardless of HIF2α dependency [40]. Abemaciclib, a CDK4/6 and PIM1 kinase inhibitor is currently being tested in phase I trial in combination with sunitinib in metastatic RCC (NCT03905889). Another compound that could represent a paradigm shift in targeted treatment is STF-62247. It has been shown to induce potent cytotoxic effects in *VHL*-deficient ccRCC cells, compared to their *VHL* wild-type counterparts [41]. The STF-62247-stimulated synthetic lethality occurs in a HIF-independent manner through autophagy; however, the mechanistic links between *VHL* and autophagy are incompletely understood [42].

### 4.4. Targeting Cancer Immune Evasion

Tumor cells interact with the immune system in a process called immunoediting, which consists of three phases: elimination, equilibrium, and escape [43]. Most of the tumor cells are destroyed in the first phase. Cells that cannot be eliminated enter the equilibrium in which they are selected through immune cell exhaustion and resistance to immune detection [44]. It is the longest of the three phases and, in ccRCC, manifests as a modest clonal expansion right after the 3p loss. The evolutionary pressure of immune predation may eventually lead to the development of mechanisms to escape immune responses. From that moment, malignant growth proceeds unrestrained. The ultimate goal of immunotherapy is to permanently reverse immune evasion strategies.

Recent phase III clinical trials led to the use of three immunotherapy-based combinations, including pembrolizumab, ipilimumab, and nivolumab, as a front-line for ccRCC [45]. These agents are highly effective, with a few patients achieving a durable complete response. The ongoing phase III clinical trials are currently testing different combinations of a checkpoint inhibitor plus a tyrosine kinase inhibitor (NCT02811861, NCT03937219) or IL-2 derivate (NCT03729245). Earlier phase studies are evaluating the potential of combining PD-1/PD-L1 inhibitors and antibodies directed against LAG-3 (NCT02996110, NCT03849469), TIM-3 (NCT02608268), or ICOS (NCT03693612, NCT03829501). An alternative approach is represented by the use of different cytokines (NCT02799095, NCT03063762) or personalized cancer vaccines (NCT03633110, NCT02950766).

ITH plays an essential role in shaping antitumor immune responses [43,44]. The highly heterogeneous tumors presumably escape immune surveillance because the reactive neoantigens undergo ‘dilution’ within the tumor, thereby leading to weaker antitumor immunity.

How do specific genomic features of ccRCC influence the clinical benefit from immunotherapy is under investigation. While tumor mutational burden (TMB) potentially increases ITH [46], a small study on 25 metastatic ccRCCs failed to confirm the association between TMB and response to immunotherapeutics [47]. Miao and colleagues found that truncating mutations in *PBRM1* were associated with significantly extended progression-free survival (PFS) and OS of patients with metastatic ccRCC treated with immune checkpoint inhibitors [48]. The underlying mechanism is probably related to increased sensitivity to T-cell-mediated cytotoxicity of *PBRM1*-mutant tumor cells [49]. This association was confirmed in an independent ccRCC cohort by a post hoc analysis of the CheckMate 025 randomized phase III study [50]. On the other hand, the exploratory analyses from JAVELIN Renal 101 and CheckMate 214 do not support this hypothesis [51,52]. The discrepant results are presumably due to the different populations studied, such as treatment-naïve versus VEGF-refractory [53].

### 4.5. Modulating Genomic Instability

Genomic instability of cancer cells drives genetic diversity required for the natural selection of adaptive traits, but there is a threshold beyond which cells cannot replicate successfully [54]. Hence, it is tempting to alter (increase or decrease) the frequency of mutations within the cancer genome.

RCC is characterized by a moderate level of genomic instability and the absence of mutations in canonical DNA damage response (DDR) genes, such as *RAD9*, *BRCA1,* or *TP53* [6,49]. As a result, RCC patients are commonly unresponsive to DNA-damaging therapies, such as chemo- or radiotherapy. For that reason, reducing genetic instability could be a more suitable approach. It can be achieved, among others, by constitutive activation of the transforming growth factor β (TGF-β) axis. TGF-β has been shown to inhibit DNA double-strand breaks (DSB) repair mechanisms to heighten the genetic diversity and adaptability of cancer cells [55]. In ccRCC cell cultures, TGF-β enhances proliferative capacity and promotes metastatic growth [56]. Early phase Ib clinical trial (NCT00356460) investigated the use of a monoclonal antibody against TGF-β fresolimumab in RCC patients and showed preliminary evidence of antitumor activity [57].

On the contrary, particular ccRCC driver genes do influence DDR and there is preclinical evidence to support the poly(ADP)-ribose polymerase (PARP) inhibition in *VHL*- or *BAP1*-mutated ccRCC [54,58,59]. Moreover, cells harboring *SETD2* mutation undergo synthetic lethal interaction with WEE1 blockade due to the depletion of nucleotide pools [60]. AZD1775, an experimental inhibitor of WEE1, is currently being evaluated for patients with *SETD2*-deficient tumors, including RCC (NCT03284385).

### 4.6. Evolutionary Herding

The tumor is less likely to be resistant to multiple drugs simultaneously, hence the combination therapy allows for the extermination of resistant cells before the emergence of further adaptive mechanisms. However, the use of two or more drugs simultaneously is strictly limited by the toxicity to normal tissues.

While checkpoint inhibitor and the antiangiogenic combination is a standard of care for metastatic ccRCC, there is a significant overlap in the toxicity profile of these drugs, with diarrhea, hypertension, and hepatotoxicity being among the most commonly presented [55,61]. These and other adverse effects may all contribute to treatment discontinuation or dose reduction. Moreover, there is frequently a need for additional medications, such as loperamide secondary to axitinib or high-dose corticosteroids for autoimmune colitis and hepatitis in case of checkpoint inhibitors. Then, drug–drug interactions become even harder to predict. Despite the toxicity issue, in most cases, cancer cells eventually develop multidrug resistance.

Any biological adaptation often involves trade-offs. In cancers, the cost of one resistance mechanism is likely to induce a population to be sensitive to an alternative therapy [62]. Evolutionary herding exploits this weakness by administering a combination of drugs in a particular order which enables to control the tumor cell population. When a second drug is administered, the clonal structure of the population is different from the start, and this may lead to enhanced sensitivity, or even complete tumor regression [63].

Since evolutionary herding alters the cellular composition of the tumor microenvironment, collateral drug sensitivity is likely to be persistent. Furthermore, this strategy is hardly influenced by stochastic perturbations and cell plasticity [64]. Acar et al. recently designed an experimental approach, in which evolution can be tightly controlled, monitored, and altered using drugs. It allows estimating evolutionary trade-offs and evaluating the effectiveness of patient-specific evolutionary herding strategies [65]. The suitability of evolutionary herding in RCC has not been tested yet.

## 5. Therapeutic Implications

As previously described, seven evolutionary trajectories can be distributed into four groups depending on the tumor’s genomic characteristics, evolution mode, and clinical course. We suspect that each group has its own unique weaknesses that could be exploited. In Figure 1, we demonstrate the predicted effectiveness of evolution-targeted strategies against particular evolutionary trajectories of ccRCC.

While the benefit of upfront CN strictly depends on life expectancy, this procedure should be considered especially for Group 1 and, to a lesser extent, Group 2. Similarly, adaptive therapy that aims to enforce a stable tumor burden is expected to be highly effective against indolent cancers. Tumors from Group 1, in which *VHL* mutation is the sole driver event, are the best candidates for targeting trunk mutations. In Group 2, there is a limited number of trunk mutations, and this approach is still reasonable. As a general rule, ITH diminishes immune responses but tumors harboring *PBRM1* mutations (Group 2) could be highly vulnerable to immunotherapeutic agents. The predictive value of *PBRM1* mutation, however, is under debate and requires further investigation. Finally, decreased wGII is an indicator of a favorable response to immunotherapy, supporting its use in Group 1. In Figure 2, we illustrate how modulating genomic instability may affect ccRCC fitness. We suggest decreasing genomic instability before the loss of 9p or 14q, which represents the acquisition of metastatic competence. This approach is particularly attractive in Group 1, characterized by low wGII. In contrast, Groups 3 and 4, due to high wGII and a punctuated evolution pattern, are expected to respond to increasing genomic instability. Modulating genomic instability in Group 2 could be unsuitable because of high wGII and branched mode of evolution. Evolutionary herding aims to decrease ITH with each subsequent therapy. Hence, it should be considered in Groups 1 and 3. This strategy may also be adequate in Group 2 due to its indolent nature in comparison to Group 4.

## 6. Future Directions

Sequencing data obtained from spatial biopsies enable one to infer the phylogenetic tree structure and, in ccRCC, estimate the evolution trajectory. As a general rule, trunk alterations are found in all tumor cells and represent an ancestral event, while other modifications constitute the branches. The more regions sampled, the more branches will be found. In low-ITH cases, four biopsies would reflect the subclonal alteration with 75% accuracy. The gain in driver detection per additional sampling declines after eight, which is usually still not enough in cases with *PBRM1* mutation [14]. While molecular profiling of multiple specimens is not practical in the setting of clinical practice, the analysis of circulating tumor DNA (ctDNA) obtained from liquid biopsy represents a feasible alternative. Analysis of ctDNA enables identification of both clonal and subclonal tumor-specific mutations with high sensitivity and specificity, with detection rates comparable with those of traditional biopsies [61,62,63,64,65,66]. Furthermore, ctDNA has a relatively short half-life (approximately 2 h), allowing for the evaluation of tumor changes in real-time [67]. Finally, as minimally invasive, liquid biopsy eliminates the morbidity associated with the serial sampling of tumors. While qualitative and quantitative analyses of ctDNA have been extensively performed in RCC patients [68], liquid biopsy has not yet been used to capture RCC evolutionary trajectories.

The discovery of alternative evolutionary trajectories of RCC will provide a better insight into the underlying mechanisms of drug resistance. Some of these mechanisms may be closely related to geographic and environmental factors since patients from different regions have different genetic backgrounds and are exposed to different carcinogens. Huang et al. identified mutational signatures and SCNAs specific to Chinese or Japanese ccRCC patients [13]. In the first group, the alterations could be due to exposure to aristolochic acid, a common ingredient in many Chinese herbs [69]. The cause of unique genetic alterations in the Japanese cohort remains unexplained.

Novel techniques to perform an in-depth analysis of datasets, as well as larger-scale studies, will greatly expand our knowledge on the development of RCC. Recently, an original computational method, CONETT (CONserved Evolutionary Trajectories in Tumors), enabled the detection of three additional directions of evolution among ccRCCs [70]. Two of them terminate with a sequence alteration in gene *KDM5C* and one in *TSC1*. The clinical significance of these findings is yet to be determined.

Identification of hidden evolutionary patterns is made possible by artificial intelligence (AI). Caravagna and colleagues devised a machine-learning method called repeated evolution in cancer (REVOLVER), which allows to overcome the stochastic effects of cancer evolution and information noise [71]. This technique uses transfer learning (TL) to achieve reproducible disease prognosis based on next-generation sequencing (NGS) count data [70,71,72,73]. As a result, it is possible to classify patients on the basis of how their tumor evolved, with implications for the anticipation of disease progression.

According to the NCCN Guidelines for Kidney Cancer, molecular profiling does not influence decision-making. The ongoing phase 2 clinical trials, A-PREDICT (NCT01693822) and ADAPTeR (NCT02446860), incorporate a multiregional sampling of metastatic RCC prior to and during therapy to evaluate biomarkers of treatment response. Whether evolutionary trajectories could reflect the effectiveness of a particular anti-RCC strategy, remains to be elucidated.

## 7. Conclusions

Many diseases are intimately tied to our evolutionary and genetic heritage. With our better understanding of these conditions, we gradually acquire the evolutionary perspective, which turns out necessary for both prevention and treatment [74,75]. In contrast to organismal evolution, human cancers are subjected to similar initial conditions and follow a limited range of possible evolutionary trajectories. Therefore, the repetitive nature of cancer evolution may prove to be its greatest weakness.

Genomic characterization is currently paving the way for clinical decision-making in RCC. The problem of exceptional ITH could be minimized by multiregion biopsy or liquid biopsy. These tools not only provide insights into cancer genetic architecture but also allow the measurement of clonal evolution. Recent studies resolved the evolutionary features and subtypes underpinning the diverse clinical phenotypes of ccRCC. In this review, we have summarized recent advances in the treatment of ccRCC and demonstrated how each strategy could be improved from the evolutionary perspective. Since there are only a few deterministic evolutionary trajectories in ccRCC, it appears feasible to use them as potential biomarkers for guiding intervention and surveillance. We believe that the presented patient stratification could help predict future steps of malignant progression, thereby informing optimal and personalized clinical decisions.

## Figures and Tables

**Figure 1 cancers-12-03300-f001:**
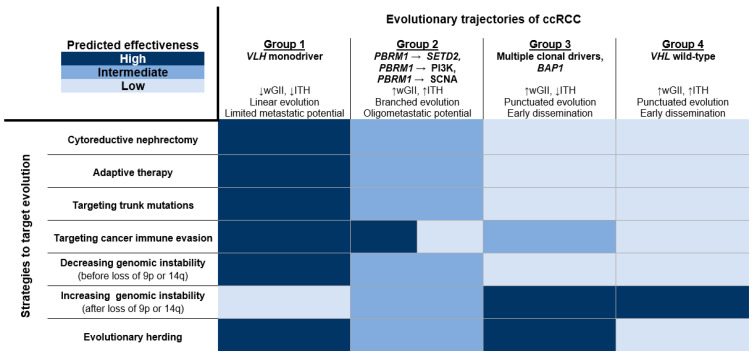
Predicted effectiveness of evolution-targeted strategies against particular evolutionary trajectories of clear cell renal cell carcinoma (ccRCC). Seven deterministic evolutionary trajectories are classified into four groups in terms of tumor’s genomic characteristics, evolution mode, and clinical course. Loss of 9p or 14q represents the acquisition of metastatic competence. There are conflicting results regarding *PBRM1* mutation as a predictive biomarker of response to immunotherapy. The figure is based on assumptions about tumor biology and therapeutic options. ccRCC, clear cell renal cell carcinoma; wGII, weighted genome integrity index; ITH, intratumor heterogeneity.

**Figure 2 cancers-12-03300-f002:**
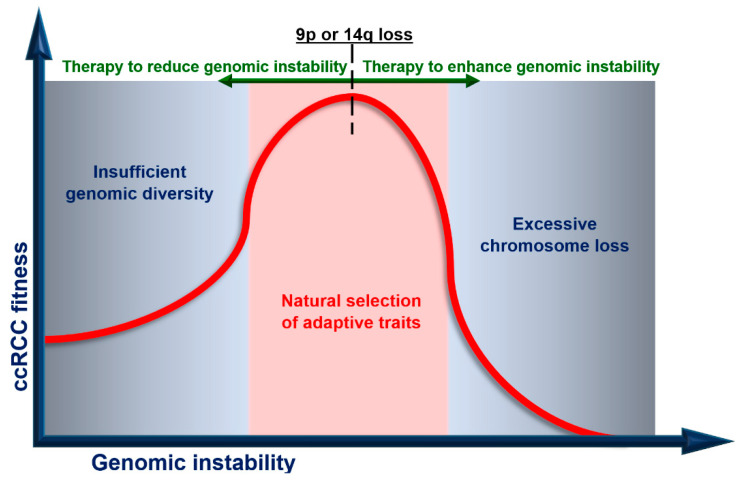
Modulating genomic instability to reduce ccRCC fitness. ccRCC fitness (vertical axis) is plotted against genomic instability (horizontal axis). There is an optimum range of genomic instability, in which ccRCC evolves. 9p or 14q loss represents the acquisition of metastatic competence and is a point of no return. Before this point is reached, decreasing genomic instability slows down cancer evolution. Once 9p or 14q is lost, increasing genome instability triggers extensive DNA damage and cell death.

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
