# Peer review of "Targeting the Deterministic Evolutionary Trajectories of Clear Cell Renal Cell Carcinoma"

_cancers, 2020, doi:10.3390/cancers12113300_

Round 1

Reviewer 1 Report

The summaries of evolution of ccRCC and the systemic therapies are comprehensive and mostly accurate. However, there are concerns that need to be addressed.

  1. On page 4, "2/1" and ""4/2" need to be defined;
  2. On page 4, the inactivation of VHL is mostly caused by somatic mutations. There is also promoter hypermethylation that reduces expression, but it is. a minor mechanism;
  3. The most significant targets of VHL loss are the production of VEGF and PDGF. Tumor angiogenesis has been the major target for over a decade, and the inhibitors of this process have been the standard of care before immunotherapy;
  4. On page 5, a synthetic lethality treatment with VHL loss by CDK4/6 inhibition (Science Signaling  01 Oct 2019: Vol. 12, Issue 601, ) should be quoted since it is under clinical trial;
  5. On page 5, targeting trunk mutation should mention progress made on the HIF2alpha antagonist. Invariably VHL loss will lead to HIF2alpha activation, and it is thought to be the core driver, if not the only one;
  6. On page 5, the sensitivity to immunotherapy conferred by PBRM1 mutations is not without dispute. There are more than one publications stating that they cannot find such a correlation. To be accurate these publications should also be cited and the statement modified;
  7. On page 7 the using of the evolution trajectory as biomarker to guide therapeutic choice is highly speculative. There is no evidence showing that evolutionary herding would work in ccRCC. Targeting trunk mutations could still work in Groups 2 and 3 if the additional mutations reduce the efficacy of therapies targeting the trunk mutations, and no evidence so far indicate that is the case. In general, the language should be toned down to state that the figure is based on reasonable speculations based on our understanding of the biology and the systemic therapies, but they await testing in the real world;
  8. If the liquid biopsy was performed in ccRCC which can detect relevant mutations and can be used to construct evolutionary trajectories, please cite it. Otherwise please state that this is goal instead of a reality.

Author Response

Comments from Reviewer #1:

The summaries of evolution of ccRCC and the systemic therapies are comprehensive and mostly accurate. However, there are concerns that need to be addressed.

  1. On page 4, "2/1" and ""4/2" need to be defined;

Response: We have explained the meaning of "2/1" and "4/2" schedules in the second paragraph of section 4.2. Adaptive therapy (page 4, lines 168-171).

  1. On page 4, the inactivation of VHL is mostly caused by somatic mutations. There is also promoter hypermethylation that reduces expression, but it is. a minor mechanism;

Response: We have added this information in the first paragraph of section 4.3. Targeting trunk mutations (page 4, lines 182-184).

  1. The most significant targets of VHL loss are the production of VEGF and PDGF. Tumor angiogenesis has been the major target for over a decade, and the inhibitors of this process have been the standard of care before immunotherapy;

Response: We have revised this sentence as suggested (page 5, lines 194-195).

  1. On page 5, a synthetic lethality treatment with VHL loss by CDK4/6 inhibition (Science Signaling  01 Oct 2019: Vol. 12, Issue 601, ) should be quoted since it is under clinical trial;

Response: Thank you or pointing this out. We have included the information on CDK4/6 inhibition and the clinical trial in the third paragraph of section 4.3. Targeting trunk mutations (page 5, lines 207-210).

  1. On page 5, targeting trunk mutation should mention progress made on the HIF2alpha antagonist. Invariably VHL loss will lead to HIF2alpha activation, and it is thought to be the core driver, if not the only one;

Response: We have described the progress on HIF2alpha inhibition in the first paragraph of section 4.3. Targeting trunk mutations (page 5, lines 194-200).

  1. On page 5, the sensitivity to immunotherapy conferred by PBRM1 mutations is not without dispute. There are more than one publications stating that they cannot find such a correlation. To be accurate these publications should also be cited and the statement modified;

Response: We are grateful for this comment as it points to an important rationale of this work. In fourth paragraph of section 4.4. Targeting cancer immune evasion, we have cited these publications and presented the possible reasons for the discrepancies (page 6, lines 265-268). We have modified Figure 1 and figure legend accordingly. Finally, we have revised the second paragraph of section 5. Therapeutic implications (page 7, lines 236-238).

  1. On page 7 the using of the evolution trajectory as biomarker to guide therapeutic choice is highly speculative. There is no evidence showing that evolutionary herding would work in ccRCC. Targeting trunk mutations could still work in Groups 2 and 3 if the additional mutations reduce the efficacy of therapies targeting the trunk mutations, and no evidence so far indicate that is the case. In general, the language should be toned down to state that the figure is based on reasonable speculations based on our understanding of the biology and the systemic therapies, but they await testing in the real world;

Response: Using of the evolution trajectory as biomarker of response to treatment is indeed speculative. In figure legend, we have clearly stated that Figure 1 is based on assumptions about the tumor biology and therapeutic options.

We agree that there is no evidence showing the effectiveness of evolutionary herding in ccRCC. At the end of section 4.6. Evolutionary herding, we have explained that this approach is yet to be tested (page 6, line 313).

We expect that targeting trunk mutations would be less effective in group 3 than in group 2 due to punctuated evolutionary pattern and tendency to disseminate early.

  1. If the liquid biopsy was performed in ccRCC which can detect relevant mutations and can be used to construct evolutionary trajectories, please cite it. Otherwise please state that this is goal instead of a reality.

Response: Thank you for this comment. Estimating the ccRCC evolutionary trajectory with liquid biopsy remains a goal to be achieved. We have modified the first paragraph of section 6. Future directions (page 8, lines 376-378).

Reviewer 2 Report

The review focuses on different genetic forms of clear cell renal cell carcinoma and how our increasing understanding of biology of these different forms and the evolutionary trajectories that lead to these tumours could be exploited for therapeutic purposes. The conventional one size fits all therapeutic approach is not adequate and there are currently no predictive genetic markers that can be used to better stratify patient therapy. This review raises ideas and discussion points that will stimulate further interest in the field to try to address this issue.

 the article is thought-provoking and encourages a different view of clear cell renal cell carcinoma, moving away from seeing the disease as one tumour entity and instead as a series of related, yet genetically diverse diseases with therapeutic implications. The article is clear and easy to read but would benefit from editing by an english native speaker to correct a few grammatical errors. My recommendation “accept with minor revision” refers to this grammatical correction, rather than sceintific additions. the authors have done a good job of posing novel hypotheses and making suggestions for therapeutic strategies that might be implemented for each tumour type. They put these suggestions in the context of our newly increased understanding of the biological features of the different genetic subtypes of tumours. The authors address the main question posed.

Author Response

Comments from Reviewer #2:

The review focuses on different genetic forms of clear cell renal cell carcinoma and how our increasing understanding of biology of these different forms and the evolutionary trajectories that lead to these tumours could be exploited for therapeutic purposes. The conventional one size fits all therapeutic approach is not adequate and there are currently no predictive genetic markers that can be used to better stratify patient therapy. This review raises ideas and discussion points that will stimulate further interest in the field to try to address this issue.

the article is thought-provoking and encourages a different view of clear cell renal cell carcinoma, moving away from seeing the disease as one tumour entity and instead as a series of related, yet genetically diverse diseases with therapeutic implications. The article is clear and easy to read but would benefit from editing by an english native speaker to correct a few grammatical errors. My recommendation “accept with minor revision” refers to this grammatical correction, rather than sceintific additions. the authors have done a good job of posing novel hypotheses and making suggestions for therapeutic strategies that might be implemented for each tumour type. They put these suggestions in the context of our newly increased understanding of the biological features of the different genetic subtypes of tumours. The authors address the main question posed.

Response: We regret there were problems with the English. The manuscript has been carefully revised by a native English speaker.